# Effects of High Temperatures on the Performance of Carbon Fiber Reinforced Polymer (CFRP) Composite Cables Protected with Fire-Retardant Materials

**DOI:** 10.3390/ma15134696

**Published:** 2022-07-04

**Authors:** Ping Zhuge, Guocheng Tao, Bing Wang, Zhiyu Jie, Zihua Zhang

**Affiliations:** 1School of Civil and Environmental Engineering, Ningbo University, Ningbo 315211, China; zhugeping@nbu.edu.cn (P.Z.); taoguocheng1995@163.com (G.T.); jiezhiyu@nbu.edu.cn (Z.J.); zhangzihua@nbu.edu.cn (Z.Z.); 2National Engineering and Research Center for Mountainous Highways, Chongqing 400067, China; 3Chongqing WanQiao Communication-Tech Co., Ltd., Chongqing 400067, China

**Keywords:** carbon fiber reinforced polymer (CFRP), fire-retardant materials, fire resistance time, temperature rise model

## Abstract

In this study, the safe critical temperature that can be tolerated by CFRP tendons under normal working conditions was derived through tensile tests at room and high temperatures. Next, the times required to reach a safe critical temperature for CFRP cables protected with different types of fire-retardant materials of various thicknesses were determined through fire resistance tests, Finally, fitting the surface of the finite element simulation results allowed the establishment of the temperature rise calculation model of CFRP tendons under the protection of fire-retardant materials. The results showed that 300 °C can be regarded as the safe critical temperature. Both high-silica needled felt and ceramic fiber felt exhibited high fireproof performance. With an increase in the thickness of the fire-retardant material, the time for the CFRP tendon to reach the inflection point of the heating rate increased, and the safe fire resistance time increased exponentially. According to the HC temperature rise curve, the fire resistance time of CFRP tendons protected by 24 mm thick high-silica needled felt was 45 min, and that for CFRP tendons protected by 24 mm thick ceramic fiber felt was 39.5 min. Under the action of fire corresponding to the hydrocarbon temperature rise model, the safe fire resistance time of CFRP tendons protected by 45 mm high-silica needled felt or 50 mm ceramic fiber felt was more than 2 h, sufficient to meet the specification. The proposed model of fire resistance performance enables the determination of the thickness of the fire resistance material required to obtain different degrees of fire resistance for CFRP cables for structural use.

## 1. Introduction

Carbon fiber reinforced polymer (CFRP) cables are lightweight and have high strength, fatigue resistance, corrosion resistance, and high seismic performance [1,2,3,4]. Carbon fiber materials are widely used [5,6,7,8,9,10,11,12] to strengthen buildings and bridges. In bridges with a cable support system, CFRP cables can be substituted for steel cables, thus reducing the weight of the bridges, decreasing the scale of the substructure, and allowing increased bridge span, reducing the overall bridge cost and the technical challenges of construction [13,14]. However, CFRP cables increase fire risk during construction and operation. The tendons in CFRP cables are composed of carbon fiber precursors and resin matrixes. The resin matrixes soften when exposed to high temperatures due to weak high temperature resistance, resulting in rapid decay and degradation of the overall mechanical performance of the tendons at high temperatures, endangering the overall safety of the bridge structure [15,16].

The mechanical performance of tendons in CFRP cables exposed to high temperatures has been studied. Wang et al. [17,18,19] tested the tensile strength of CFRP tendons with a diameter of 9.5 mm at different temperatures and found that 350°C was the critical temperature for these materials. At temperatures below 350 °C, the tensile strength of CFRP tendons decreased with decreasing temperature. At temperatures higher than 350 °C, the material strength decreased sharply and the performance became unstable. Yu and Kodur [20] studied the mechanical performance of CFRP tendons used to consolidate reinforced concrete beams by embedding at 20–600 °C. They observed a slight decrease in the mechanical performance of carbon fiber tendons at 200 °C, but a 50% change at 300 °C, which was attributed to the use of carbon fiber material matrix epoxy resin glue. The epoxy resin glue begins to decompose when the temperature exceeds 300 °C, and the carbon fibers lose most of their bearing capacity at 400 °C. Hamed et al. [21] conducted an experimental study on the axial tensile performance of CFRP tendons subjected to temperature increasing from 30 to 200 °C over 3 h. They found that when the treatment temperature was lower than the glass transition temperature Tg = 110 °C, there was negligible degradation of tensile strength of the tendons, but at treatment temperature higher than Tg, the axial tensile strength of the tendons decreased significantly. Sumida et al. [22] tested the ultimate tensile strength of CFRP tendons subjected to a temperature increasing from 50–280 °C for 0.5 h and found that at treatment temperature lower than 260 °C, the axial tensile strength and elastic modulus of the tendons decreased only slightly. Zhou et al. [23] conducted tensile tests on CFRP tendons with a diameter of 8 mm at high temperatures, and the test results showed less degradation of tensile strength of CFRP tendons at treatment temperatures lower than Tg = 126 °C than at room temperature. Additionally, the degradation law of elastic modulus was found to be similar to that of tensile strength. Fang et al. [24] conducted axial tensile tests on CFRP tendons with a diameter of 4.17 mm exposed to 30 °C, 100 °C, and 200 °C for 3 h, and found little changes in ultimate tensile strength and elastic modulus of CFRP tendons with exposure to 100 °C which hardly decreased, but the ultimate tensile strength of CFRP tendons decreased by 28.5% and the elastic modulus decreased by 9.9% after exposure to 200 °C for 3 h compared with those exposed to 30 °C. Mohammad et al. [25] conducted tensile tests on CFRP tendons with diameters of 4 and 5 mm after treatment at temperatures from 25 °C to 800 °C and found that the ultimate tensile strength of CFRP tendons decreased by approximately 44%. A control group test was carried out on CFRP tendons coated with nitrogen-based intumescent coatings. When the test temperature ranged from 350 °C to 600 °C, the nitrogen-based intumescent coatings had a certain protective effect on the CFRP tendons, reducing the loss of tensile strength by approximately 5%. Esmaeil et al. [26] conducted pull-out tests on CFRP tendons with diameters of 4 and 5 mm embedded with concrete at both ends. The ultimate tensile strength of CFRP tendons hardly decreased at 80 °C, decreased by approximately 52% at 400 °C, and only 25% of the ultimate strength of CFRP tendons remained at 800 °C, with almost complete loss of bearing capacity. Xu et al. [27] applied a pre-stress 0.19–0.57 times the ultimate tensile stress to CFRP tendons with a diameter of 5 mm at a temperature of 20–600 °C, and then performed an axial tensile test after 20 min at the target temperature in the pre-stressing environment. The test results showed that at the same ambient temperature, the high pre-stress state aggravated the damage of the temperature to the ultimate bearing capacity of the CFRP tendons. Phi et al. [28] established a model of CFRP with fire-retardant material through ABAQUS for finite element simulation and used this model to predict fire resistance performance. Joao et al. [29] studied concrete beams strengthened with CFRP sheets using calcium silicate board and vermiculite/perlite cement-based mortar layer as a fire protection system. They found greater separation time between CFRP sheets and concrete beams under the protection of a 40 mm thick fire protection system than that when a 25 mm thick system was used, based on finite element simulation and experimental results. Mohamed et al. [30] proposed a finite element simulation method to evaluate the heat transfer performance of insulated steel columns with consideration of heat transfer, heat convection, and heat radiation. The calculation results were consistent with those obtained by the finite element simulation method.

It is clear that high temperatures can significantly impact the mechanical performance of CFRP tendons. Therefore, there is a significant need to protect CFRP cables using fire-retardant materials. To obtain effective protection, it is necessary to study the effects of high temperatures on the performance of modified CFRP cables. However, few studies have analyzed the mechanical performance of carbon fiber cables containing fire-retardant materials.

Given this lack of information, the goal of this work was to study the fire resistance performance at different temperatures of tendons in carbon fiber cables wrapped with fire-retardant materials. First, the safe critical temperature that can be endured by the CFRP tendons under normal operation conditions was determined by a comparative test of the CFRP tendons under normal and high temperatures. Second, the fire resistance time for the CFRP bar to reach the safe critical temperature under the protection of different types of fire-retardant materials of various thicknesses was determined through fire resistance tests. The results of these tests were used for subsequent numerical simulation of fire resistance performance. ABAQUS finite element software was used to establish a numerical simulation model of CFRP tendons wrapped with fire-retardant materials. The times for CFRP tendons protected with different materials of various thicknesses to reach the safe critical temperature were determined based on the fire heating curve of bridge engineering, and a corresponding calculation model was established. The model provides reference for the fire protection of CFRP cables for engineering.

## 2. Tensile Tests of CFRP Tendons at High Temperatures

### 2.1. Design and Fabrication of Specimens

CFRP tendons were purchased from Zhongao Carbon Fiber Technology Co., Ltd., Suzhou, China, with a sand-coated surface. The tensile test was performed on CFRP tendons with a diameter of 9 mm, length of the free end of 600 mm, and length of anchorage at both ends of 150 mm. Strand tapered anchorage devices were used for anchoring and the specific sizes of the tested specimens are shown in Figure 1. The strand tapered anchorage device is composed of an anchor cylinder, clips, and an aluminum sleeve. The assembly process of the anchoring system is shown in Figure 2.

### 2.2. Design of Test Conditions

Tests were carried out on a Servo-Hydraulic Universal Testing Machine(which is provided by Jinan Xinguang Testing Machine Manufacturing Co., Ltd., Jinan, China) equipped with a high temperature furnace. The loading system is shown in Figure 3. Previous reports [17,18,19] showed that the mechanical properties of CFRP bars decrease significantly at about 350 °C. Therefore, a total of five temperature conditions were designed for the test, room temperature (25 °C), 250 °C, 300 °C, 350 °C, and 400 °C, corresponding to samples NT-1, HT-1, HT-2, HT-3, and HT-4, as shown in Table 1.

As shown in Figure 3a, the tensile force was controlled during the test by a universal servo testing machine and the temperature was controlled using the external control box GW-1200B of the heating furnace located in the middle of the instrument (as shown in Figure 3b). To prevent the effects of high temperatures on anchorage performance, asbestos was placed at both ends of the heating furnace during the test to prevent heat escape. Specimen NT-1 was subjected to a tensile test at a speed of 1.5 kN/s until failure. Specimens HT-1-HT-4 were first stretched to the effective stress of 700 MPa (approximately 44.5 kN) corresponding to the conditions of practical engineering after considering the loss of pre-stress, the load was held, and then the free section of the CFRP tendon was heated at 50 °C/min to the target temperature before holding at that temperature for 2 h (heating curve shown in Figure 3c). The specimens were continuously stretched at a rate of 1.5 kN/s until failure. In this way, the tensile force-displacement curves of CFRP tendons were obtained under different working conditions and at normal and high temperatures. The rate of change of the elastic modulus was determined for samples before and after the action of high temperatures.

### 2.3. Test Results and Analysis

The ultimate tensile strength under load and heating in a heating furnace for 2 h was tested for all samples except for Specimen NT-1. Figure 4 shows the tensile force–displacement for the operating conditions of each test. The slopes of the tensile force and displacement of the CFRP tendons in the linear relationship were roughly consistent, as shown. There were slight fluctuations in the lateral direction for the tension–displacement relationships of the specimens when the load was held for 2 h under heating conditions, with significant lateral displacement for Specimen HT-4 at 400 °C. Additionally, as shown in the enlarged region, the load of CFRP tendon can still be stably maintained at about 44.5 kN under the action of high temperature, with no early fracture. When loaded again after 2 h of high temperature furnace heating, each specimen exhibited a significantly changed tensile force–displacement slope, which decreased with the increase in temperature. The slope of Specimen HT-4 was reduced to 57.9% of the initial value when it was subjected to a high temperature load at 400 °C for 2 h. The ultimate bearing capacity of each specimen also decreased with increased operating temperature, and the ultimate bearing capacity of Specimen HT-4 was only 59.3% of that of NT-1. The mechanical properties of CFRP bars change greatly under the action of 400 °C because the epoxy resin adhesive in CFRP bars liquefies and loses its bonding force under the action of high temperature.

Figure 5 shows the failure diagram for CFRP tendons under various test conditions. As shown, high temperatures caused different degrees of damage to the CFRP tendons during the heating process. The CFRP tendons in Specimen NT-1 showed an explosive fracture state, which fully exerted the tensile strength of the CFRP tendons. The surface of the CFRP tendons in Specimen HT-1 was charred black, and this charred residue could be detached from the surface. However, when the CFRP tendons were bent, the stiffness of the CFRP tendons did not change significantly at room temperature, suggesting that the carbon fiber precursors can basically maintain their original strength after exposure to 250 °C for two hours. The behavior of CFRP tendons in Specimen HT-2 was similar to that of HT-1, with a charred black surface and no significant decrease in stiffness. However, observation of the failure section of the specimen revealed that more of the epoxy resin glue inside the tendons loses strength due to the increase in temperature. Specimen HT-3 had multiple sites in the middle part of the CFRP tendons which expanded significantly after exposure to 350 °C and the carbon precursors became looser, suggesting that most of the colloids lost their cementation capacity after the high temperature exposure. On bending the CFRP tendons, the carbon precursor became a little soft, and the stiffness decreased significantly. The CFRP tendons in HT-4 finally fractured and failed in a hair-like form, with complete detachment of the carbon precursors and loss of cementation capacity of the interior colloids. The carbon precursors were soft, with a significant drop in strength and stiffness.

The slope was calculated at two points in the linearly increasing part of the tensile force–displacement diagram for each specimen, before heating (around 40 kN) and after heating (around 60 kN). The ratio of the two slopes was the rate of change of elastic modulus before and after heating of the specimen. The rate of change of elastic modulus of the CFRP tendons before and after heating of the five specimens and the test results of the ultimate tensile strength are shown in Table 2.

It can be seen from Table 2 that the elastic modulus and ultimate tensile strength of CFRP tendons were reduced by varying degrees after 2 h of heating at a high temperature. The reduction coefficient decreased with the increase in the test temperature, in roughly a linear relationship. The specific relationship diagrams are shown in Figure 6 and Figure 7.

Based on the above test results, CFRP tendons could maintain a pre-tension of 700 MPa after exposure to a high temperature of 400 °C for 2 h, but the ultimate tensile strength and elastic modulus of CFRP tendons decreased by 40.7% and 41.6%, respectively. The appropriate setting of safety coefficients must consider uncertainties such as the differences between the mechanical performance, test values, and design values of the material and their values in practical use. Based on the fitting curves of the tensile strength reduction coefficient and elastic modulus reduction coefficient and the temperatures in Figure 6 and Figure 7, the tensile strength and elastic modulus of CFRP tendons exposed to a high temperature of 300 °C for 2 h were equivalent to more than 80% of the values at lower temperature. As a result, 300 °C can be regarded as the safe critical temperature of the carbon tendons during normal use. Thus, if proper fire-retardant material can protect the CFRP tendons against temperatures exceeding 300 °C that could occur under the action of fire, the structural performance of the CFRP tendons can be effectively guaranteed.

## 3. Fire Resistance Assessment of Fire-Retardant Material of CFRP Tendons

Protection of CFRP cables against fire requires flexible fire-retardant materials that are easily curled. In this study, high-silica needled felt and ceramic fiber felt were tested as fire-retardant materials for protection of CFRP tendons. The fire-resistant performance of these materials as a fire-resistant layer was tested at different thicknesses. Investigation of the protective effects of the two materials provides data support to establish a numerical simulation calculation method for fire protection design. Fire intensity is mainly determined by the temperature in the heating furnace, as controlled by the control box. Fire resistance performance is expressed as the time when the CFRP tendon reaches the critical safe temperature as measured by the thermocouple located on the surface of the CFRP tendon.

### 3.1. Fire-Retardant Materials

Two types of fire-retardant materials, high-silica needled felt and ceramic fiber felt, were tested at a thickness of 6 mm, as shown in Figure 8. High-silica needled felt is made of chopped strands of high-silica glass fiber. When carded into cotton and needle punched, high-silica needled felt has a small pore size and a high porosity. This high-temperature resistant filter material has reasonable structure and high performance, as shown in Figure 8a. Ceramic fiber felt has excellent fire resistance, with advantages of low thermal conductivity, light weight, low heat capacity, high chemical stability, strong corrosion resistance, excellent thermal stability, non-flammable in case of naked fire, and suitable for long-term use at 1000 °C. Ceramic fiber felt is often used for high-temperature thermal insulation in aerospace, petrochemical, and military equipment, as shown in Figure 8b. The two fireproof materials were purchased from Zhejiang Oka Refractories Co., Ltd., Huzhou, China.

### 3.2. Test Scheme

To assess fire resistance performance, four groups of CFRP tendons were protected with high-silica needled felt and ceramic fiber felt with different thicknesses (see Table 3 for the specific test conditions and see Figure 9 for the specimens). The GW-1200B high-temperature heating furnace and the control box were used to control the heating rate at 50 °C/min and the tension force was controlled by the universal servo testing machine. Considering that the maximum temperature of about 1100 °C is reached typically in actual fire accidents, such as would occur with an explosion of an oil tank truck on a bridge structure, 1100 °C was used as the target temperature of the heating furnace. The surface temperature of the CFRP tendons was determined using a thermocouple and a temperature-measuring instrument. Both ends of the heating furnace were blocked with fireproof asbestos to prevent heat escape. The detailed design is presented in Figure 10.

During the test, tensile stress of 700 MPa (approximately 44.5 kN) was applied on the CFRP bar by the universal servo testing machine. The heating furnace applied a high temperature to the CFRP bar under the protection of the fireproof material, with a target of 1100 °C. The temperature change on the surface of the CFRP bar was monitored by a thermocouple between the fireproof layer and the CFRP bar, allowing derivation of the heating curve of the CFRP bar upon exposure to high temperature.

### 3.3. Test Results and Analysis

Figure 11 shows the heating curve of the surface of CFRP tendons wrapped with high-silica needled felt of different thicknesses. The black line in the figure indicates the temperature in the high-temperature heating furnace, the blue line indicates the surface temperature of the CFRP bar, the red solid point represents the inflection point of the heating rate (the inflection point of the heating rate is the zero position of the third derivative of the heating curve), and the red dotted line Tc denotes the safe critical temperature of 300 °C. According to the heating curve, the temperature in the heating furnace reached 1100 °C at 22 min of increased temperature, while the temperature of the protected CFRP tendons increased very slowly in the initial stage. However, with the increase in time, the rate of temperature increase of the CFRP tendons increased. Using a thicker layer of high-silica needled felt, the time required for the CFRP tendons to reach the critical safe temperature of 300 °C increased accordingly. For 24 mm thick high-silica needled felt, the CFRP tendons reached the critical safe temperature of 300 °C in 45 min. A longer time was required to reach the inflection point of the heating rate with the increase in thickness of the fire-retardant material, and the heating rate after the inflection point decreased with the increase in thickness. The heating rate after the inflection point was significantly higher than the rate before the inflection point. The heating rate of CFRP tendons was always 50 °C/min lower than the heating rate of the high-temperature furnace due to the protection of the fire-retardant materials. At a thickness of 6 mm, the heating rate reached 43.8 °C/min. It is worth noting that the CFRP tendons were able to maintain a tensile stress of 700 MPa during the entire heating process.

As shown in Figure 12, the ceramic fiber felt performed similarly to the high-silica needled felt in the fire performance test. The time for the CFRP bar to reach the safe critical temperature of 300 °C increased with increased thickness of ceramic fiber. For ceramic fiber 24 mm thick, the CFRP bar reached the safe critical temperature of 300 °C in 39 min. Based on a parallel comparison of the two fire-retardant materials, the inflection point of the heating rate of the high-silica needled felt was significantly later than that of the ceramic fiber felt. Thus, the time for CFRP tendons to reach the safe critical temperature under the protection of the high-silica needled felt was significantly longer than for those protected by ceramic fiber felt. As with those protected by high-silica needled felt, CFRP tendons protected with ceramic fiber felt maintained a tensile stress of 700 MPa throughout the whole heating process.

Figure 13 shows the time required for CFRP tendons protected with different thicknesses of the two materials to reach the critical safe temperature of 300 °C. As shown in Figure 13, increasing thickness of the fire-retardant material was linearly correlated with an increase in the time for the CFRP bar to reach the safe critical temperature. Parallel comparison of the two materials revealed slightly better fire performance of the high-silica needled felt compared to that of the ceramic fiber felt. At the same thickness, the time for the CFRP tendons protected with high-silica needled felt to reach the critical safe temperature was approximately 5 min longer than that for CFRP tendons protected with ceramic fiber felt.

## 4. Numerical Simulation of the Fire Resistance of CFRP Tendons Coated with Fire-Retardant Materials

### 4.1. Finite Element Simulation Calculation

Numerical simulation calculation of the fire resistance performance of CFRP tendons coated with fire-retardant materials was performed, and the calculation results were compared with the above experimental results to verify the validity of the simulation. In the finite element calculation model, the heating model is the same as the heating process of the high temperature furnace used in the experiment test. To simplify the analysis, the following assumptions were made: (1) All fireproof materials are homogenous, have the same thermal conductivity in all directions, and have stable physical performance during the test; (2) there is proper contact between the fireproof materials; (3) no heat was generated inside the model, and only external heating was considered; (4) the temperature change of the specimen in the height direction was ignored; (5) only the temperature field of the model was considered, for uncoupled heat transfer analysis.

The thermal parameters for the high-silica needled felt and ceramic fiber felt used in the finite element model were obtained from the manufacturer or derived from previous experiments. The high-silica needle felt has a density of 120 kg/m^3^ and a specific heat capacity of 800 J/(kg·°C), with thermal conductivity as shown in Table 4. The ceramic fiber felt has a density of 100 kg/m^3^ and a specific heat capacity of 600 J/(kg·°C), with thermal parameters as shown in Table 5.

In the model, the outer fire-retardant material wraps internal CFRP tendons 9 mm in diameter, for a total length of the fire-retardant material of 300 mm. Numerical simulation was performed to determine the internal temperature field distribution of the fire-retardant material, so establishment of a CFRP bar model was not necessary. A space of 9 mm diameter was reserved inside the fire-retardant material. The model is shown in Figure 14. The 300 mm length section in the middle of the model is the cylindrical fireproof layer rolled from the fireproof material, with sections 10 mm in length on both ends that are fully thermally insulated end plugs to prevent the entry of heat from the ends of the fire-resistant material.

For the test, the entire heating section of the specimen was completely placed in the GWM-1200B high-temperature heating furnace. In the samples, the fire-retardant material of the specimen was homogeneous in texture, and tightly enclosed the CFRP tendons in a cylindrical shape, so it was assumed that the temperature of the specimen does not vary with height and angle during the test. In other words, the temperature change only occurs in the diameter direction, with uniform distribution in the other directions. The heating curve of the external temperature of the fire-retardant material increased at a rate of 50 °C/min at 50% power of the high-temperature heating furnace and then the temperature was held when it increased to 1100 °C.

Using ABAQUS software, the heat transfer performance of fire-retardant materials of CFRP tendons was simulated by the finite element method. The three-dimensional heat transfer model was adopted for the finite element model, using the 8-node linear heat transfer hexahedral element DC3D8 element. By defining the boundary conditions, the temperature field Amp-1 (temperature rise curve is shown in Figure 14) was applied to the outer surface of the model, and the predefined temperature field of the fire-retardant materials was set as 20 °C. There are two forms of heat transfer: heat transfer and heat convection. Heat transfer occurs from the outer layer to the inner side of the fire-retardant materials, as defined in the material properties, and heat convection is the heat exchange between the external surface of the model and the air. The surface of the fire-retardant materials was selected and the corresponding heat exchange coefficient was set as 1500 W/m^2^·K. In this finite element simulation, four models were built for different thickness of fire-retardant materials. Figure 14 shows the finite element model for materials with 6 mm thickness.

When the finite element model calculation was completed, the temperature change data of the element nodes on the middle inner surface of the fire-retardant materials with time were collected and used to generate a cloud diagram of the cross-sectional temperature distribution of the fire-retardant materials when the inner surface temperature was 300 °C, as shown in Figure 15 and Figure 16.

For establishment of the finite element model, the density, specific heat capacity, and thermal conductivity of the fire-retardant materials of CFRP tendons were obtained from the data given by the manufacturer. However, the thermal conductivity given by the manufacturer is a certain value, but in practice, the thermal conductivity will change with the change of temperature. Thus, the thermal conductivity of the fire-retardant materials was slightly adjusted in the simulation process to improve the accuracy of the model.

### 4.2. Comparisons and Analyses of Numerical Simulation Results and Experimental Results

The numerical simulation calculation results and test results of the fire resistance of the high-silica needled felt are compared in Figure 15 below. The cloud diagram in the figure is the cross-sectional diagram of the middle section of the fire-retardant material when the inner temperature of the fire-retardant material reaches 300 °C. See the cloud diagram legend in the upper left corner. According to the figure, the heating trends of the CFRP tendons obtained by the numerical simulation and the test under the protection of the fire-retardant material essentially coincide. Table 6 presents the specific error and error rate between the fire resistance times of the two to reach the safe critical temperature of 300 °C. As shown in Table 6, the maximum error rate between the finite element simulation results and the test results was 12.69%, the minimum error rate was less than 4%, and the standard deviation was about 1. Overall, the finite element simulation results are close to the test results.

**Figure 15 materials-15-04696-f015:**
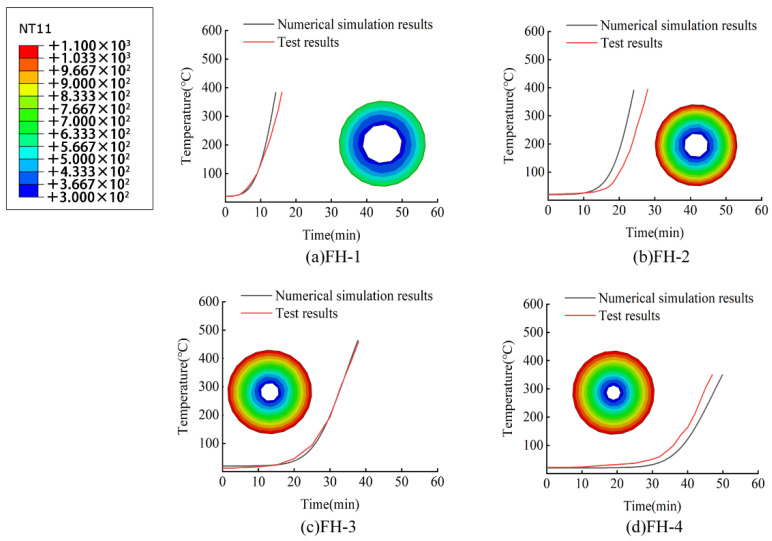
Comparison between finite element simulation and test results of fire resistance of high-silica needled felt.

The numerical simulation calculation results and test results of the fire resistance performance of the ceramic fiber felt are compared in Figure 16 below. According to the figure, the heating trends of the CFRP tendons obtained by the numerical simulation and the test under the protection of the fire-retardant material essentially coincide. Table 7 presents the specific error and error rate between the fire resistance times of the two to reach the safe critical temperature of 300 °C. As shown in Table 7, the error rate between the finite element simulation results and the test results was less than 15%, the minimum value was only 4.91%, and the standard deviation was about 1. Again, the finite element simulation results are close to the test results.

**Figure 16 materials-15-04696-f016:**
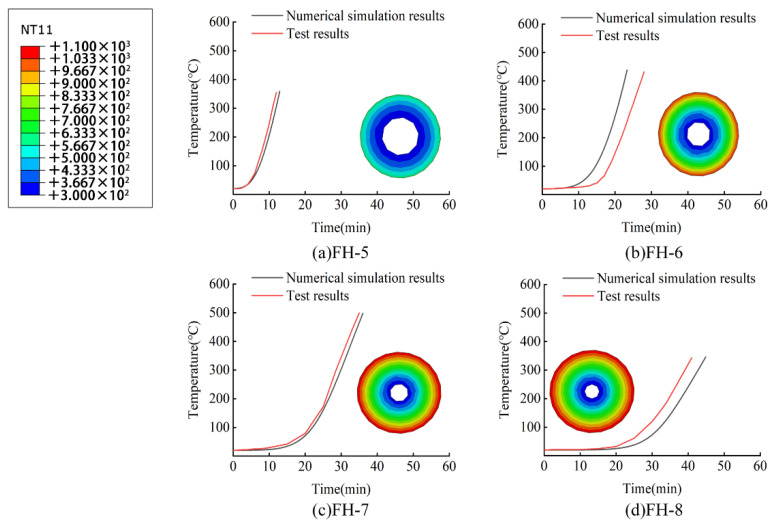
Comparison between finite element simulation and test results of fire resistance of ceramic fiber felt.

According to Table 6 and Table 7, the time for the interior of the fire-retardant material to reach the safe critical temperature in the numerical model essentially coincides with the test results. The average difference between the fire resistance simulation result and the test value is 8.09% for the high-silica needled felt and 9.94% for the ceramic fiber felt. It is thus clear that the proposed finite element numerical simulation method is reliable. The error between simulation and test may be caused by the deviation between the thermal conductivity of fireproof materials in the finite element model and the actual value—the thermal conductivity in the model is obtained by fine tuning the values given by the manufacturer.

## 5. Design Calculation of CFRP Cable Fire Protection System

### 5.1. Heating Curve of Finite Element Model

Various curves can be used to simulate fire heating, including the ISO-834 standard heating curve and the hydrocarbon (HC) heating curve [14]. The heating process is shown in Figure 17 and Figure 18.

Utilized in the design of buildings related to petrochemical and offshore oil industries in the 1980s, the HC curve describes the fire and combustion characteristics of hydrocarbons as the main fuel. The hydrocarbon (HC) curve can be used to simulate the heating curve of the fire caused by large-area leakage resulting from the collision and rupture of large flammable tanks, such as gasoline or chemical product transport tanks. The HC heating curve has a high heating rate in the early stage, with a temperature rise greater than 900 °C within 5 min, which approximates the fire heating model of flammable and explosive vehicles deflagrating on a bridge deck. Hence, this model was employed in the finite element calculations. The HC heating curve can be used in the numerical simulation analysis of the fire action of a bridge reinforced with CFRP cables [27], with the calculation formula as follows:(1)$$T=1080(1-0.325e^{-0.167t}-0.675e^{-2.5t})+T_{0}$$
where, *t* is the time, in minutes (min); *T* is the average temperature in the furnace at *t*, in degrees Celsius (°C); *T*_0_ is the initial average temperature, 20 °C. 

### 5.2. Calculation of Fire Protection Design for CFRP Cables

The calculation for the design of the fire protection system for CFRP cables was conducted using the finite element calculation method presented in Part 4 above. The nature of the CFRP material was not considered in the modeling, due to the close fitting of the fire-retardant material and the CFRP tendons. The temperature of a certain point in the innermost layer of the fire-retardant material can represent the surface temperature of the CFRP tendons. According to Section 2.3, the safe critical temperature of the CFRP cable was Tc = 300 °C. The time for the temperature of the innermost layer of the fire-retardant material to reach 300 °C was selected as the fire resistance time of the CFRP tendons, and 180 min was taken as the simulated fire resistance time.

The fire-retardant materials of high-silica needled felt and ceramic fiber felt with different thicknesses were selected for the simulation. The numerical simulation results of the internal temperature of the CFRP tendons under the action of the hydrocarbon (HC) heating curve are shown in Figure 19 and Figure 20.

Figure 19 and Figure 20 present the heating curves inside different types of fire-retardant materials with different thicknesses derived from the obtained finite element simulation results. The black dotted line at 120 min in the figure corresponds to China’s “Code for Fire Protection of Architectural Design” [31], which stipulates that the fire resistance time of bridge structures is two hours. According to Figure 19 and Figure 20, the temperature trends of protected CFRP tendons were similar for the different materials and can be roughly divided into three stages: gentle rise, rapid rise, and approaching fire peak temperature. The first and second stages are demarcated by the inflection point of the heating rate, represented by the solid red dot in the figure. In the first stage, external heat needs to be conducted into the interior of the fire-retardant material through convection, radiation, and heat conduction, and the internal temperature increased slowly. The time in the first stage increased with increasing thickness of the fire-retardant material. In the second stage, the internal temperature of the fire-retardant material rose sharply. In this stage, the heat was continuously conducted into the fire-retardant material through heat conduction. The thermal performance of the fire-retardant material changed under the action of high temperature, which accelerated the rate of heat conduction. During this stage, the heating rate of the inner CFRP tendons increased significantly, but with an increase in the thickness of the fire-retardant material, the rate slowed. In the third stage, the rate of temperature increase inside the fire-retardant material slowed, and gradually approached the peak temperature of the fire. As shown, under the protection of 45 mm thick silica needle felt or 50 mm thick ceramic fiber felt, more than two hours was required for CFRP tendons to reach the critical safe temperature of 300 °C.

To further study the relationship between fire-retardant materials with different thicknesses and the fire resistance time and heating rate, the critical data from the temperature curves in Figure 19 and Figure 20 were extracted for additional analysis. Figure 21 presents the time required for the differently enclosed carbon tendons to reach the inflection point of the heating rate. As shown, with increased thickness of the fire-retardant material, the time required for the CFRP tendons to reach the inflection point of the heating rate increased. For the same thickness, the time for the CFRP tendons protected with high-silica needled felt to reach the inflection point of the heating rate was longer than that for tendons protected with ceramic fiber felt. The high-silica needled felt exhibited better fire resistance in the initial heating stage compared to that of ceramic fiber felt. The time for the CFRP tendons to reach the inflection point of the heating rate under the protection of 60-mm thick high-silica needled felt was 169.6 min, much longer than the safe fire resistance time of 120 min.

Figure 22 presents the increase rate of the temperature of differently protected CFRP tendons at the critical temperature of 300 °C. As shown, with the increase in the thickness of the fire-retardant material, the heating rate of the CFRP tendons decreased when the temperature reached 300 °C. The decreasing rate gradually slowed with the increase in the thickness of the fire-retardant material, exhibiting non-linear variation. When the CFRP bar reached 300 °C, the heating rate of the ceramic fiber felt-protected bar was lower than that of the high-silica needled felt-protected CFRP bar. This results shows that ceramic fiber felt can provide better fire protection in this stage than high-silica needled felt.

Figure 23 presents the safe critical fire resistance time of CFRP tendons protected with fire-retardant materials of different thicknesses. As shown in the figure, with the increase in the thickness of the fire-retardant material, the fire resistance time increased exponentially. At the same time, the safe critical fire resistance of CFRP tendons protected with high-silica needled felt was significantly longer than that of CFRP tendons protected with ceramic fiber felt. When the high-silica needled felt with a thickness of only 45 mm was employed to protect the CFRP tendons, the time to reach the critical safe temperature of 300 °C was 121.7 min, longer than the two hours stipulated in the specification and therefore adequate to meet the design requirements.

Based on the simulation results, after two hours of the HC heating curve, for CFRP tendons protected with 60 mm fire-retardant material, the temperature of the tendons changed in the thickness direction inside the fire-retardant material. According to Figure 24, at the point where the CFRP tendons contact the fire-retardant material, that is, where the thickness of the fire-retardant material was 0, the temperature of the high-silica needled felt was 20.1 °C, indicating no increase. In contrast, the temperature of the ceramic fiber felt reached 58.0 °C. Thus, under the action of the 2 h HC heating curve, the 60 mm fire-retardant material protected the CFRP tendons. In addition, near the heating furnace, the temperature changes of the two materials were almost consistent, but as the temperature continued to penetrate into the interior of the fire-retardant material, the heat resistance of the high-silica needled felt was significantly superior to that of ceramic fiber felt.

To establish a uniform calculation model for fire performance, surface fitting was performed for the three fire performance parameters of the two fire-retardant materials, as shown in Figure 19 and Figure 20. Where, the time *t*-thickness of fire-retardant material *h*-temperature *T_CFRP_* is as shown in Figure 25 and Figure 26, the correlation coefficients of surface fitting for high-silica needle felt and ceramic fiber felt are R_2_ = 0.932 and R_2_ = 0.938, respectively, exhibiting a satisfactory fitting result. The uniform calculation formula for the two fire-retardant materials is as follows:(2)$$T_{CFRP}=(1000\ln t-55h)\cdot k-T_{n}$$
where, *k* is the thermal resistance coefficient of the fire-retardant material, 0.62443 for the high-silica needle felt and 0.65797 for the ceramic fiber felt; *T_n_* is the maximum temperature of the HC curve, 1100 °C.

China’s “Code for Fire Protection in Architectural Design” specifies the different requirements for various components of different structures for safe fire resistance time depending on their importance. The fire resistance performance calculation model (Formula (2)) can be used to calculate the appropriate thickness of fire resistance material required for different fire resistance times to protect CFRP cable for use in different structures.

## 6. Conclusions

The following conclusions can be drawn based on the high temperature performance of CFRP tendons, the tests of the fire resistance performance of the CFRP tendons protected by high-silica needled felt or ceramic fiber felt under the action of high temperature, the simulated calculations based on the data of the fire performance tests using ABAQUS, and the calculations of the fire performance of the protected tendons under the HC heating mode using the numerical simulation method:

(1) A relationship between the tensile strength-temperature and elastic modulus-temperature of CFRP tendons was established based on the measured high temperature performance of CFRP tendons. The tensile strength and elastic modulus of CFRP tendons decreased with increasing temperature in the range of 250–400 °C, essentially exhibiting a linear decreasing trend. After being exposed to a high temperature of 300 °C for 2 h, the CFRP bar maintained its tensile strength and elastic modulus at more than 80% of the initial values. Thus, 300 °C may be used as the critical temperature for fire protection design of carbon tendons.

(2) Based on the fire performance test, the relationship of thickness and fire resistance time was derived. The fire resistance time increased with increased thickness of the fireproof material. At the same thickness, high-silica needled felt exhibited better fire performance than that of ceramic fiber felt. The fire resistance time of CFRP tendons protected by 24 mm thick high-silica needled felt was 45 min, and that for CFRP tendons protected by 24 mm thick ceramic fiber felt was 39.5 min.

(3) Numerical simulation calculations of the fire resistance performance of CFRP tendons enclosed by high-silica needled felt or ceramic fiber felt as a fireproof layer revealed that the heating process can be roughly divided into three stages: gentle rise, rapid rise, and approaching fire peak temperature. With increasing thickness of the fire-retardant material, the time for the CFRP bar to reach the inflection point of the heating rate increased, the heating rate decreased at 300 °C, and the safe fire resistance time increased exponentially.

(4) Under exposure to fire corresponding to the hydrocarbon (HC) temperature rise model, CFRP tendons protected by 45 mm high-silica oxygen needled felt or 50 mm ceramic fiber felt exhibited safe fire resistance times of greater than two hours, as required by the specification. The fire resistance performance calculation model (Formula (2)) for the design proposed here allows determination of the appropriate thickness of the fire resistance material required for different fire resistance times for CFRP cables used in different structures.

(5) Fire-retardant materials can protect carbon fiber tendon materials but could also be used to protect other bridge components such as pre-stressed steel strands of bridges and CFRP tendon anchorages.

## Figures and Tables

**Figure 1 materials-15-04696-f001:**
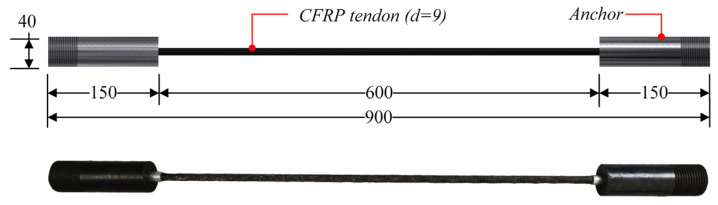
Specimen of tensile test (Unit: mm).

**Figure 2 materials-15-04696-f002:**
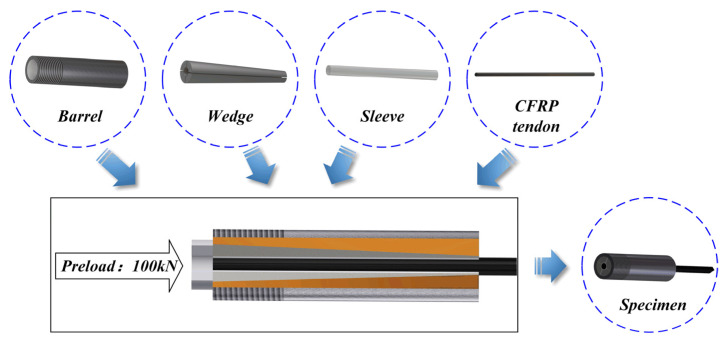
Fabrication process of the anchoring system.

**Figure 3 materials-15-04696-f003:**
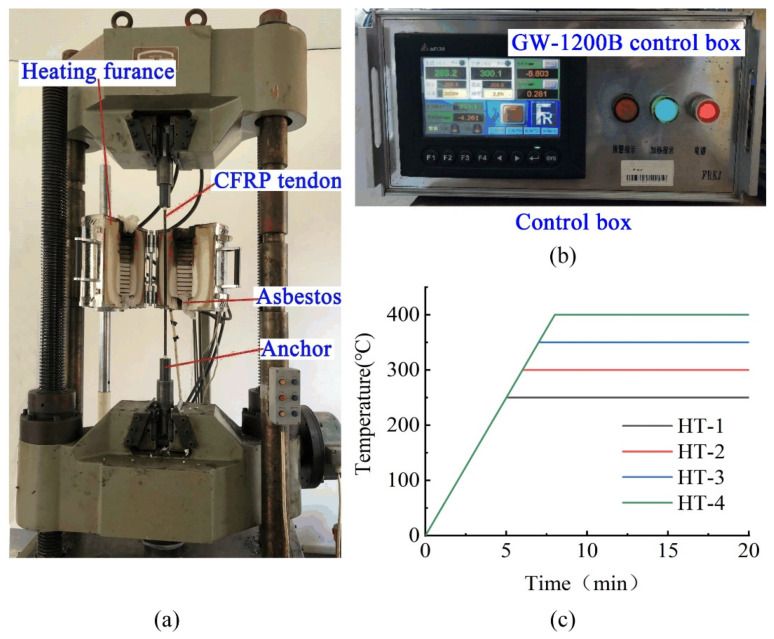
Tensile test process of CFRP tendon at high temperature. (**a**) Test process (**b**) control box (**c**) Heating curve of heating furnace.

**Figure 4 materials-15-04696-f004:**
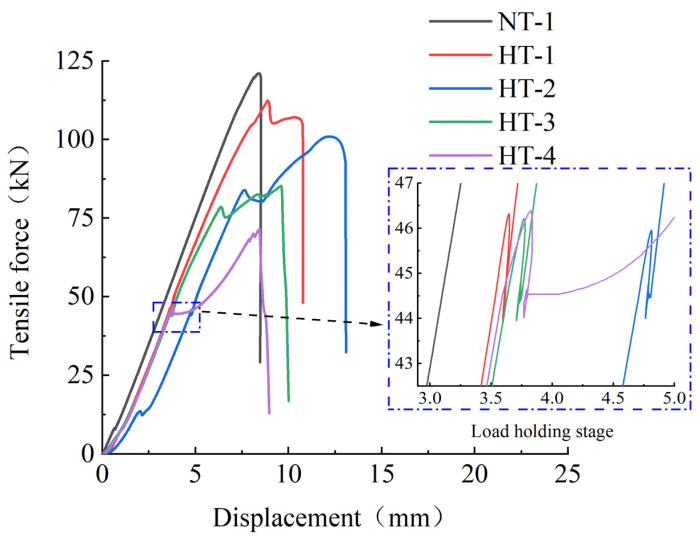
Tensile force–displacement of each test.

**Figure 5 materials-15-04696-f005:**
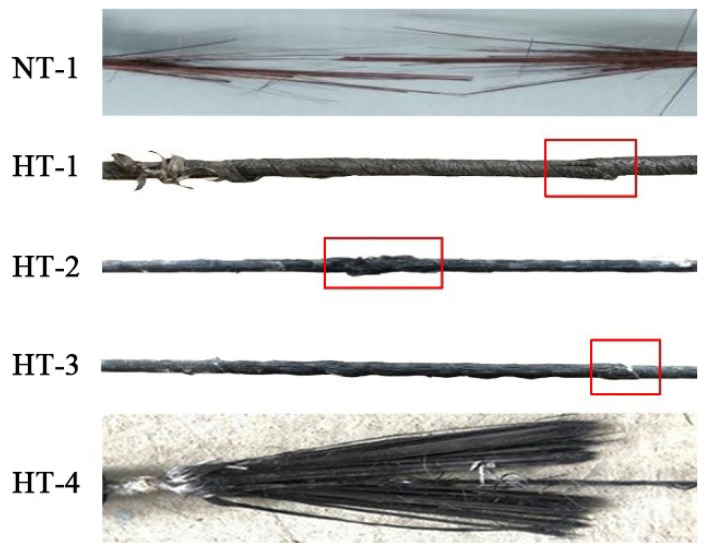
Failure mode of CFRP tendons under various test conditions.

**Figure 6 materials-15-04696-f006:**
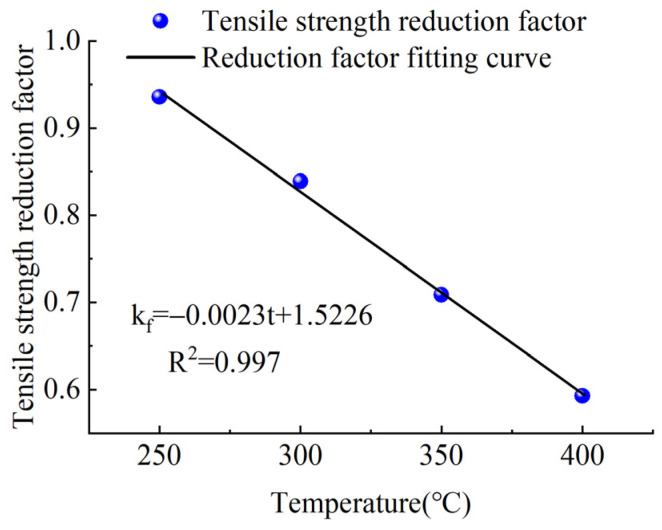
Reduction coefficient of tensile strength.

**Figure 7 materials-15-04696-f007:**
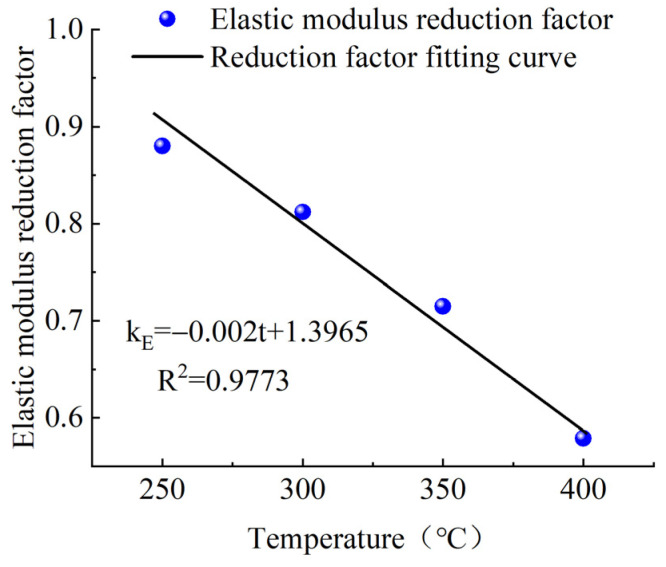
Reduction coefficient of elasticity modulus.

**Figure 8 materials-15-04696-f008:**
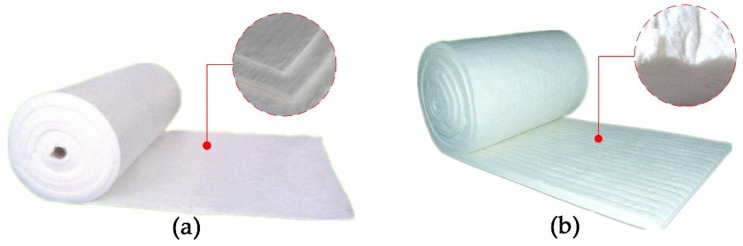
Fire-retardant materials. (**a**) High-silica needled felt, (**b**) ceramic fiber felt.

**Figure 9 materials-15-04696-f009:**
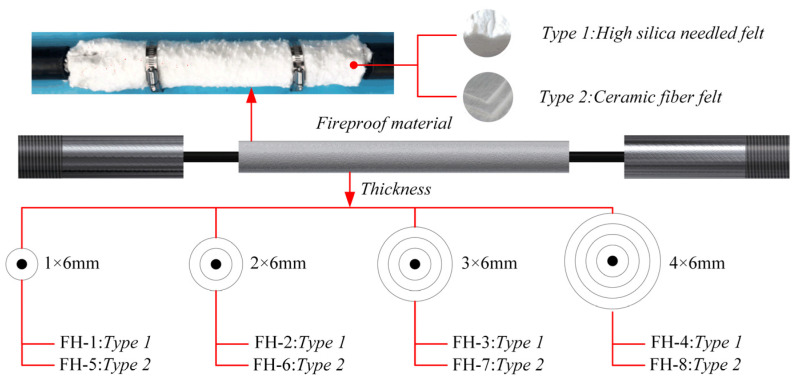
Specimen of fire resistance test.

**Figure 10 materials-15-04696-f010:**
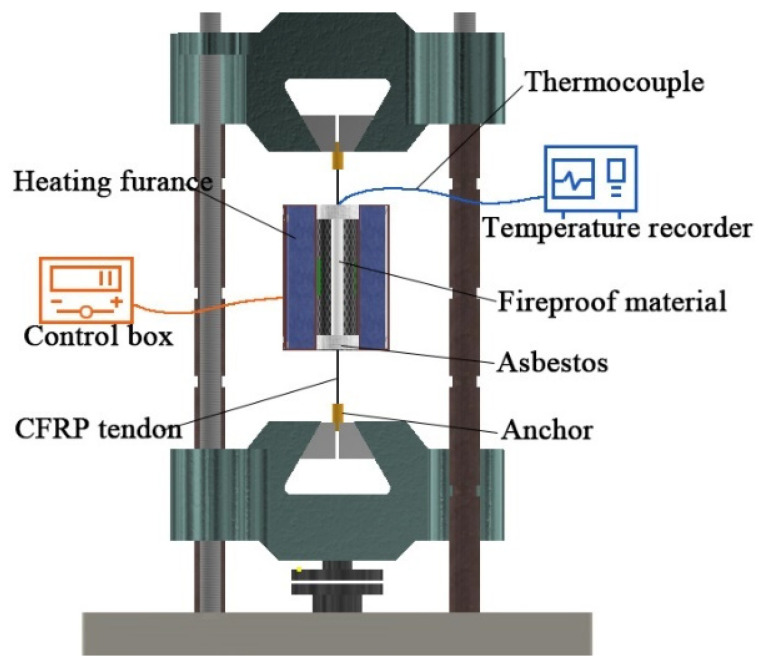
Test device of fire resistance test.

**Figure 11 materials-15-04696-f011:**
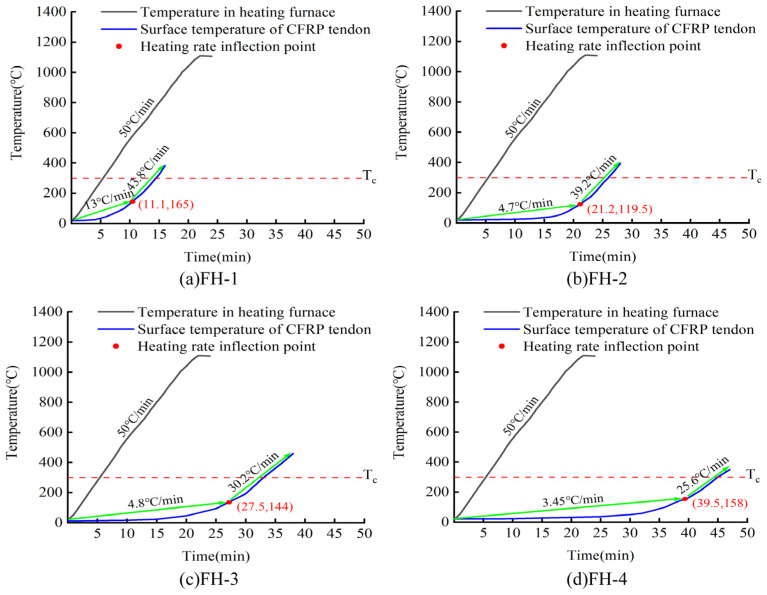
Heating curve of the surface of CFRP tendons wrapped with high-silica needled felt of different thicknesses.

**Figure 12 materials-15-04696-f012:**
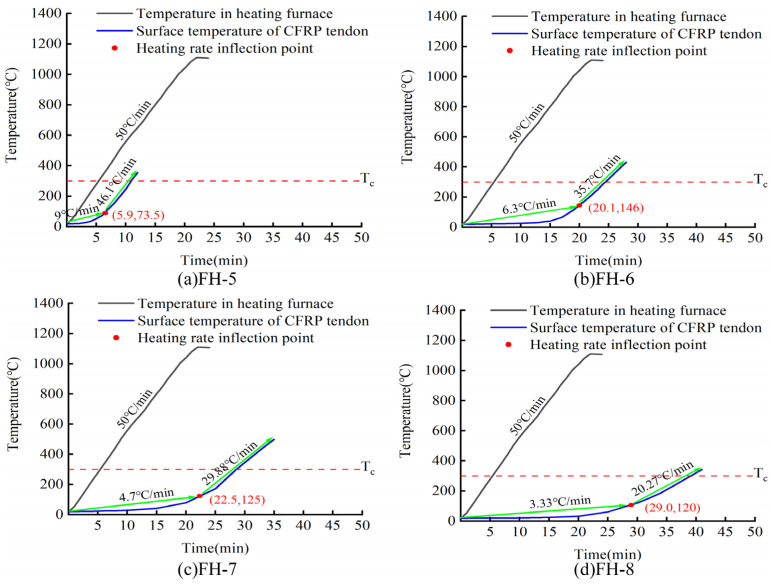
Heating curve of the surface of CFRP tendons wrapped with ceramic fiber felt of different thicknesses.

**Figure 13 materials-15-04696-f013:**
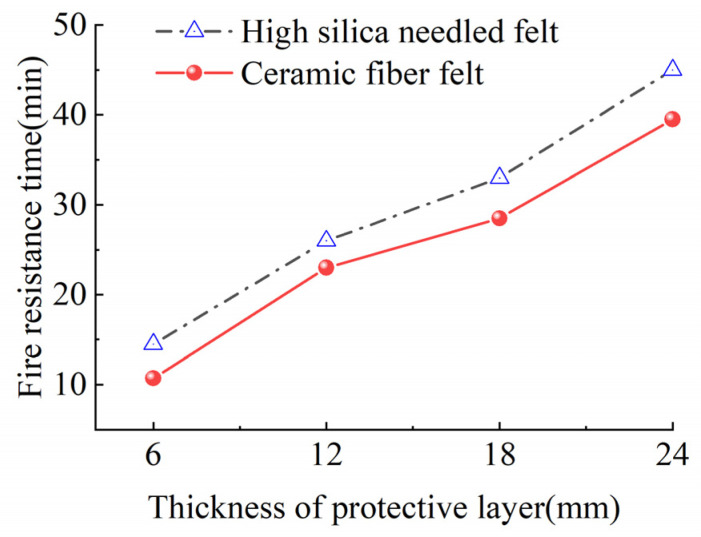
Fire resistance times under different thicknesses of protective layer.

**Figure 14 materials-15-04696-f014:**
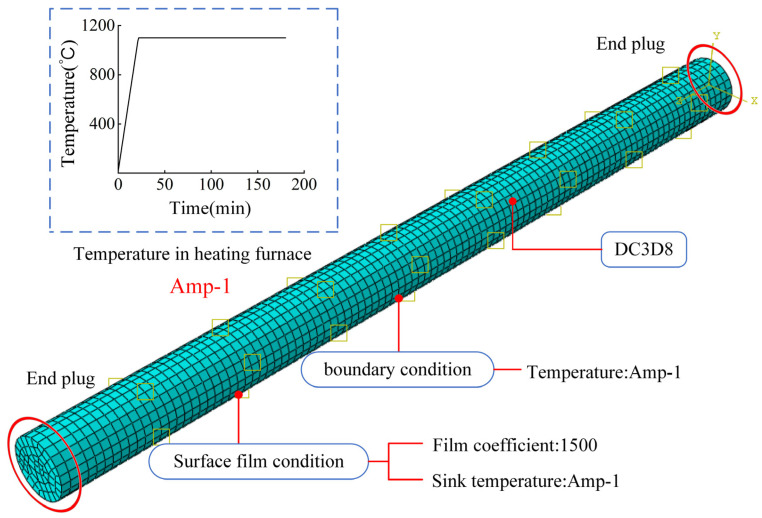
Finite element model.

**Figure 17 materials-15-04696-f017:**
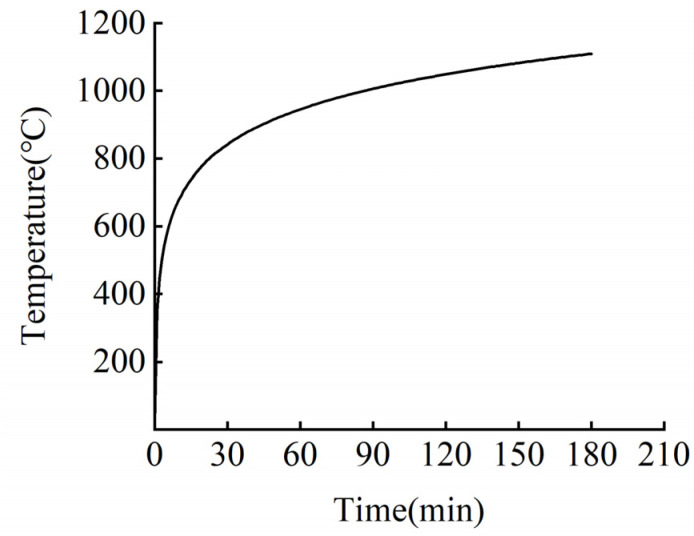
Temperature rise curve of ISO-834.

**Figure 18 materials-15-04696-f018:**
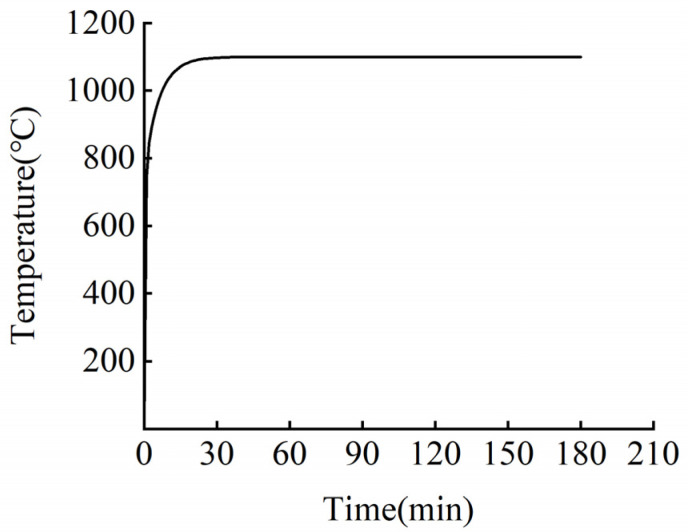
Temperature rise curve of HC.

**Figure 19 materials-15-04696-f019:**
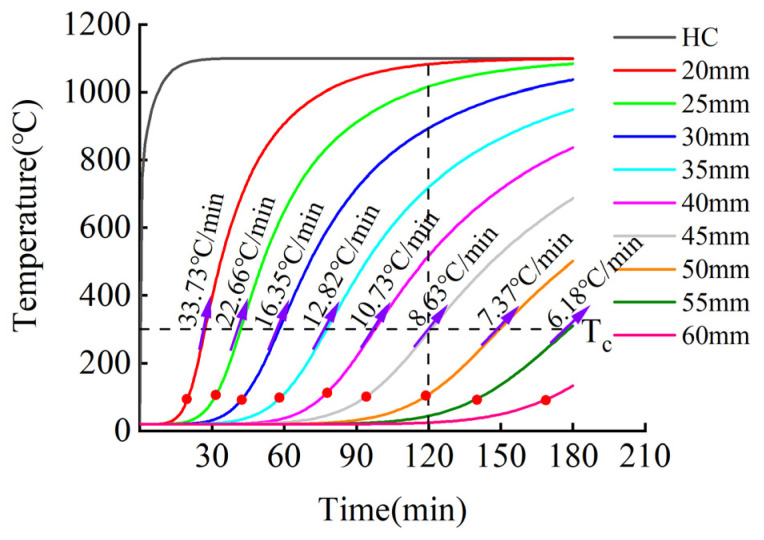
Temperature-time curve of CFRP tendons protected with high-silica needle felt.

**Figure 20 materials-15-04696-f020:**
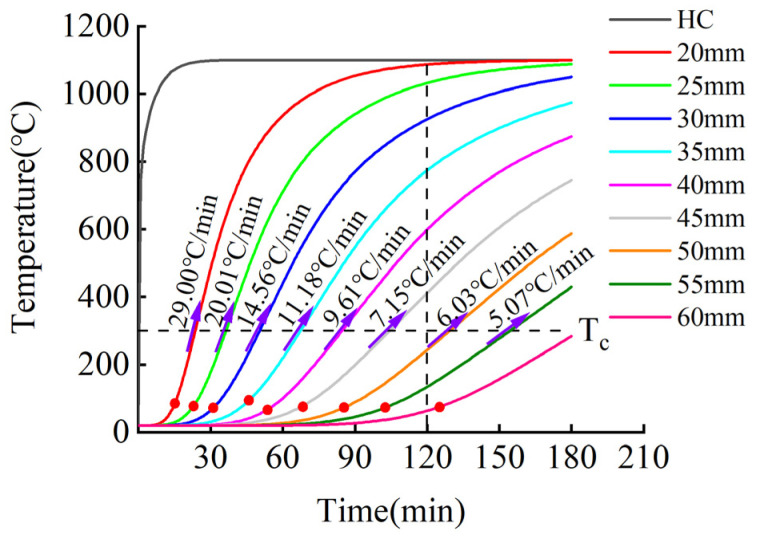
Temperature-time curve of CFRP tendons protected with ceramic fiber felt.

**Figure 21 materials-15-04696-f021:**
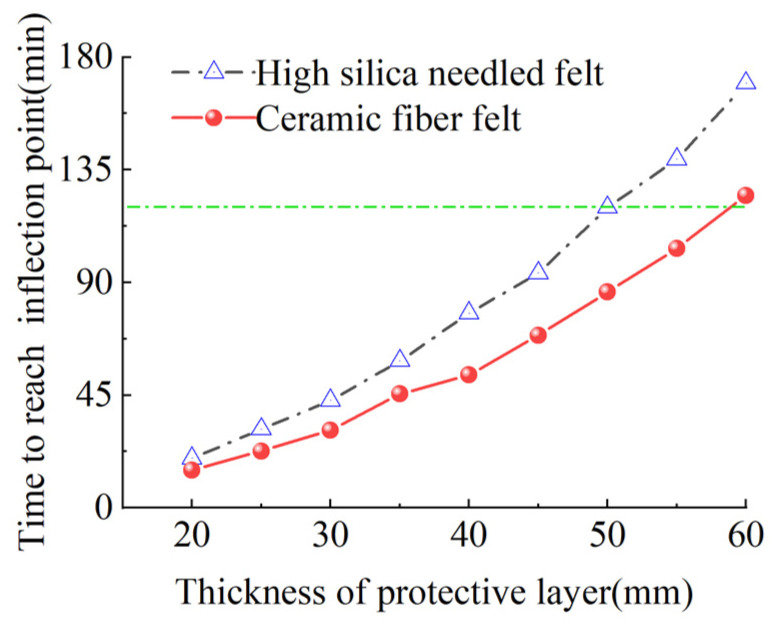
Time for heating rate to reach inflection point.

**Figure 22 materials-15-04696-f022:**
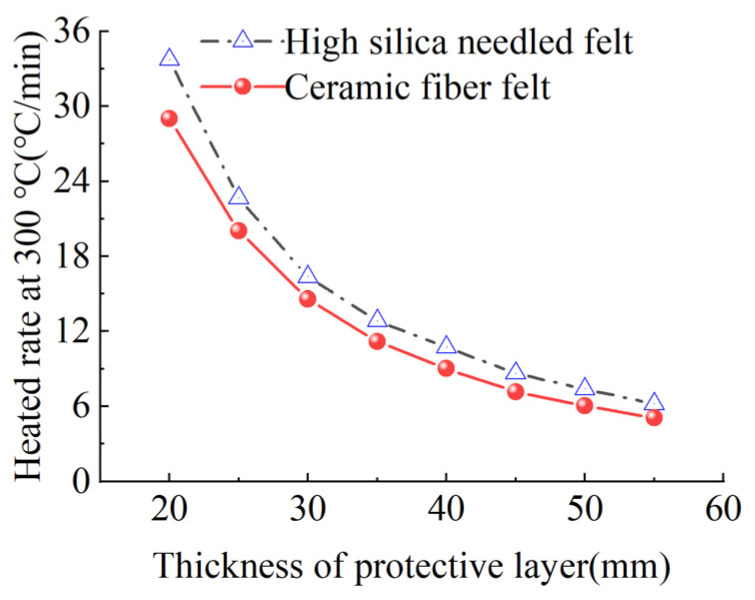
Heating rate at 300 °C.

**Figure 23 materials-15-04696-f023:**
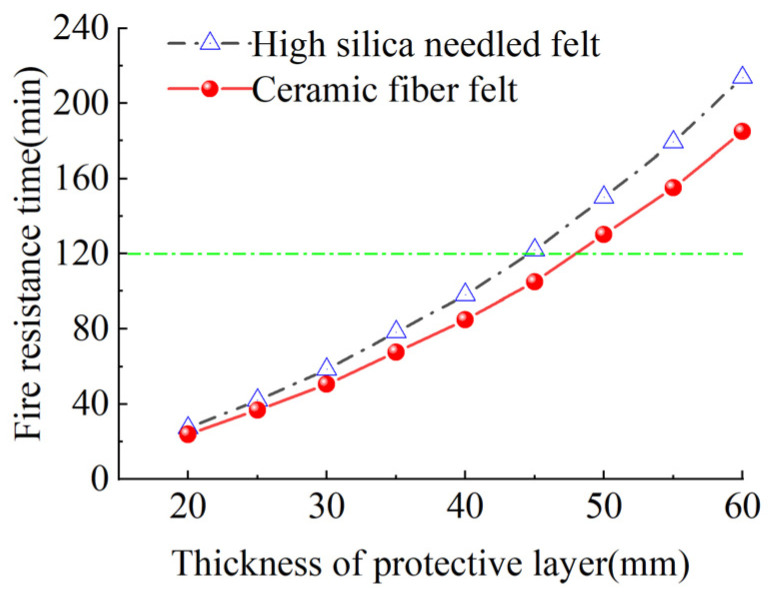
Critical duration of fire resistance duration.

**Figure 24 materials-15-04696-f024:**
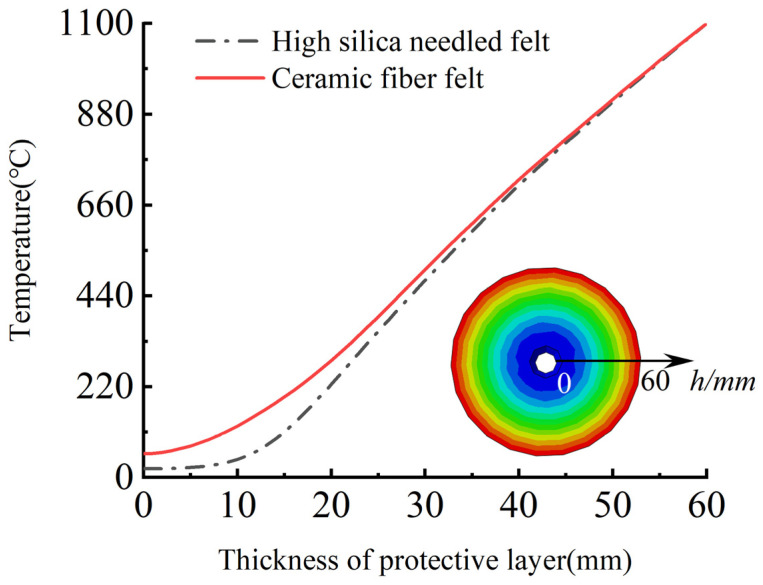
Variation of temperature along the thickness direction of fire-retardant materials during heating for 2 h.

**Figure 25 materials-15-04696-f025:**
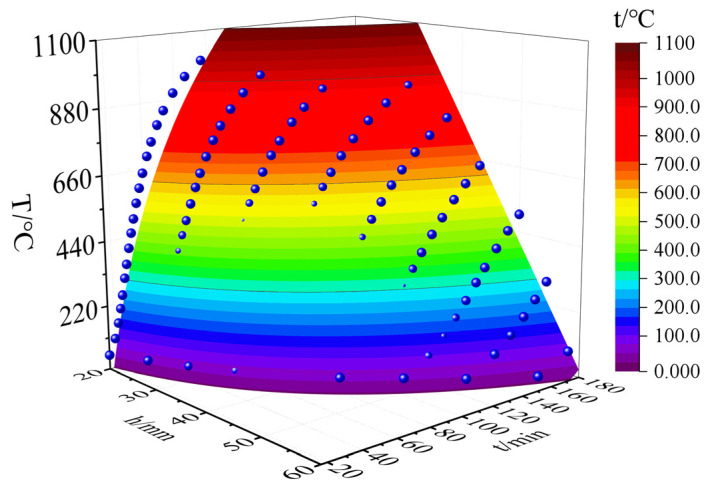
Surface fitting results of high-silica needled felt.

**Figure 26 materials-15-04696-f026:**
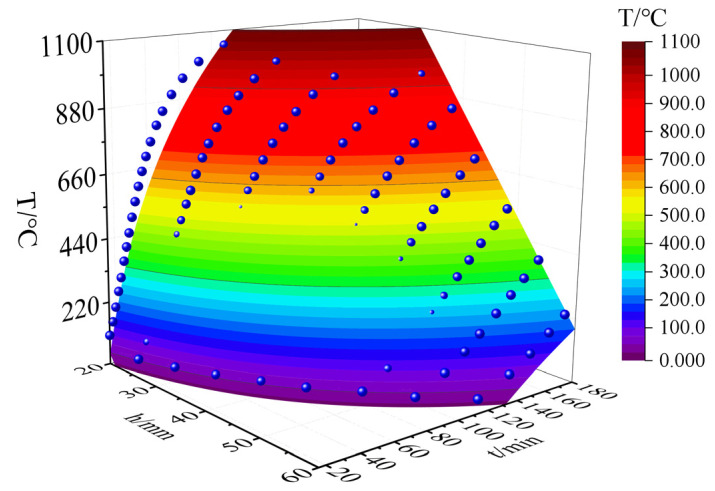
Surface fitting results of ceramic fiber felt.

**Table 1 materials-15-04696-t001:** Test conditions of tensile test specimen.

Specimen No	Temperature/°C	Diameter/mm	Loading Time/h	Preload/kN
NT-1	Ordinary temperature	9	\	\
HT-1	250	9	2	44.5
HT-2	300	9	2	44.5
HT-3	350	9	2	44.5
HT-4	400	9	2	44.5

**Table 2 materials-15-04696-t002:** Tensile test results of each specimen.

Specimen No	Tensile Force/kN	Tensile Strength/MPa	$k_{f}$	$k_{E}$
NT-1	120.3	1887	/	/
HT-1	112.3	1766	0.936	0.880
HT-2	101.0	1588	0.839	0.812
HT-3	85.3	1341	0.709	0.715
HT-4	71.3	1121	0.593	0.579

Note: *k_f_* is the reduction coefficient of the tensile strength of the carbon tendons at high temperatures and is the ratio of the tensile strength after heating to the tensile strength at room temperature; *k**_E_* is the reduction coefficient of the elastic modulus of the carbon tendons at high temperatures, which is the rate of change of the elastic modulus before and after heating.

**Table 3 materials-15-04696-t003:** Working conditions of test specimen.

Number	Fire-Retardant Materials	Thickness/mm	Preload/kN	Temperature/°C
FH-1	Type 1: High-silica needled felt	6	44.5	1100
FH-2		12	44.5	1100
FH-3		18	44.5	1100
FH-4		24	44.5	1100
FH-5	Type 2: Ceramic fiber felt	6	44.5	1100
FH-6		12	44.5	1100
FH-7		18	44.5	1100
FH-8		24	44.5	1100

**Table 4 materials-15-04696-t004:** Thermal parameters of high-silica needled felt.

Temperature/°C	Thermal Parameters W/(m·k)
20	0.05
200	0.12
400	0.22
600	0.3
800	0.38

**Table 5 materials-15-04696-t005:** Thermal parameters of ceramic fiber felt.

Temperature/°C	Thermal Parameters W/(m·k)
20	0.06
200	0.12
400	0.15
600	0.2
800	0.25

**Table 6 materials-15-04696-t006:** Errors between test and simulation results of fire resistance times of high-silica needled felt.

Specimen No	Fire-Resistance Times of Experiment/Min	Fire-Resistance Times of Finite Element Simulation/Min	Standard Deviation	Error Value/Min	Error Rate
FH-1	14.5	13.2	0.65	−1.3	−8.97%
FH-2	26.0	22.7	1.65	−3.3	12.69%
FH-3	32.0	33.2	0.6	+1.2	+3.75%
FH-4	45.0	48.1	1.55	+3.1	+6.89%

**Table 7 materials-15-04696-t007:** Errors between test and simulation results of fire resistance times of ceramic fiber felt.

Specimen No	Fire-Resistance Times of Experiment/Min	Fire-Resistance Times of Finite Element Simulation/Min	Standard Deviation	Error Value/Min	Error Rate
FH-5	10.7	11.9	0.6	+1.2	+11.21%
FH-6	24.2	20.6	1.2	−2.4	+14.8%
FH-7	28.5	29.9	0.7	+1.4	−4.91%
FH-8	39.5	43.0	1.75	+3.5	+8.86%

## Data Availability

The data used to support the findings of this study are available from the corresponding author upon request.

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
