# Peer review of "Effects of High Temperatures on the Performance of Carbon Fiber Reinforced Polymer (CFRP) Composite Cables Protected with Fire-Retardant Materials"

_materials, 2022, doi:10.3390/ma15134696_

Round 1

Reviewer 1 Report

1) In the abstract, please mention 1 or 2 lines regarding the methodology adopted.

2) In the introduction section, add some more studies conducted on finite element methods for fire protection of CFRP.

3) From where CFRP tendons were procured.

4) How fire rating and corresponding fire resistance was measured

5) How modeling was done in ABAQUS. Which models used for materials modelling. Explain in complete details regarding pre-processing and post processing of ABAQUS model used.

6) Fig. 15: model results are matching exactly with the experimental results. Can u explain how calibration was done in ABAQUS model.

7) Can u add separate section for the discussion of experimental and model results.

8) For references writing and its formatting or style, journal guidelines for writing the references need to be followed. 

Reviewer 2 Report

The reviewed work entitled "Effects of High Temperatures on the Performance of Carbon Fiber Reinforced Polymer (CFRP) Composite Cables Protected with Fire Retardant Materials" by Ping Zhuge, Guocheng Tao, Bing Wang, Zhiyu Jie, and Zihua Zhang presents the results of mechanical tests of CFRP cables in In the context of fireproof properties, the authors indicate the importance of such research works and indicate that the works so far have not touched upon this aspect. In my opinion, the conception of the work is interesting and the reviewed manuscript was prepared appropriately, however, sometimes there is no scientific discussion and attempts to explain the phenomena. Nevertheless, I have a few doubts: Why was only the temperature up to 400oC taken into account? What is the reason for such a dramatic change in the mechanical properties of the HT-4 sample (Fig. 4) In addition, fig. 4 is difficult to read and should be corrected. Please also provide a full description of the materials used in the research (type, origin, etc.) In my opinion, the peer-reviewed work, with minor changes, is suitable for publication in the Materials journal.

Reviewer 3 Report

Dear Authors

This manuscript is focused on the safety-critical temperature that can be tolerated by CFRP tendons under normal working conditions and was derived through tensile tests at room and high temperatures. Next, the times required to reach a safe critical temperature for CFRP cables protected with different types of fire-retardant materials of various thicknesses were determined through fire resistance tests, allowing modelling of the fire resistance time by finite element calculation. The results showed that 300 °C can be regarded as a safety-critical temperature. The following suggestion and comments should be taken:

1. The authors could insert more numerical data into the Abstract for enhancement of the manuscript.

2. The overall English needs to be improved. Please seek guidance from a native English speaker if possible ("the" "a", commas, plural form and others could be corrected).

3. The introduction section needs enhancement 1-3 sentences about carbon fibres/carbon nanotubes composites and their different potential applications. Please cite: (1) Materials 2022, 15(12), 4252; https://doi.org/10.3390/ma15124252 (2) Materials 2021, 14(9), 2448; https://doi.org/10.3390/ma14092448 (3) Materials 2022, 15(12), 4270; https://doi.org/10.3390/ma15124270

4. Could the authors include the standard deviation of the used methods?

5. Figure 4. Please correct this image for better quality (the inscriptions mainly in small image).

6. In table 3. Why do authors not use a higher temperature? Please explain

7. Please explain more about the error rate in table 6.

8. Figures 19 and 20. Please correct these images for better quality.

9. Figure 24. Please correct this image for better quality.

10. Authors are suggested to describe some future plans or potential applications of materials in conclusions.

Round 2

Reviewer 1 Report

Comments were incorporated. So manuscript may be accepted for possible publication.

Reviewer 2 Report

I would like to thank the authors for clearing up my doubts regarding the reviewed work. The authors made changes in the manuscript, which increase its substantive value, therefore, in my opinion, the work may be allowed for further editorial work.

Reviewer 3 Report

The authors have addressed all comments and the manuscript can be published as is.